# Role of Truncated *O*-GalNAc Glycans in Cancer Progression and Metastasis in Endocrine Cancers

**DOI:** 10.3390/cancers15133266

**Published:** 2023-06-21

**Authors:** Diluka Pinto, Rajeev Parameswaran

**Affiliations:** 1Division of Endocrine Surgery, National University Hospital, Singapore 119074, Singapore; mihindukulasooriya_diluka@nuhs.edu.sg; 2NUS Centre for Cancer Research, Yong Loo Lin School of Medicine, National University of Singapore, Singapore 117599, Singapore; 3Department of Surgery, Yong Loo Lin School of Medicine, National University of Singapore, Singapore 119228, Singapore

**Keywords:** aberrant, *O*-glycans, cancer, metastasis, endocrine

## Abstract

**Simple Summary:**

The surface of a human cell is coated by glycans, which play an important role in many of the physiological processes that include cell–cell communication, interaction, and adhesion. The aberration of the process of glycan synthesis plays a role in inflammatory conditions, tumour progression and metastasis. The process of *O*-glycosylation is complex and takes place in the Golgi apparatus, regulated by specific enzymes called *pp*GalNAc transferases. The incomplete synthesis or truncated forms of *O*-glycans such as Tn, STn, T and ST antigens are commonly seen in cancer states. The increased expression of these truncated glycans is associated with increased invasion potential, leading to metastasis and poor prognosis in a wide range of cancers. Understanding the expression of truncated glycans by cancers paves the way for targeted therapies, which have the potential to be used as serum biomarkers of disease progression and prognosis.

**Abstract:**

Glycans are an essential part of cells, playing a fundamental role in many pathophysiological processes such as cell differentiation, adhesion, motility, signal transduction, host–pathogen interactions, tumour cell invasion, and metastasis development. These glycans are also able to exert control over the changes in tumour immunogenicity, interfering with tumour-editing events and leading to immune-resistant cancer cells. The incomplete synthesis of *O*-glycans or the formation of truncated glycans such as the Tn-antigen (Thomsen nouveau; GalNAcα- Ser/Thr), its sialylated version the STn-antigen (sialyl-Tn; Neu5Acα2–6GalNAcα-Ser/Thr) and the elongated T-antigen (Thomsen–Friedenreich; Galβ1-3GalNAcα-Ser/Thr) has been shown to be associated with tumour progression and metastatic state in many human cancers. Prognosis in various human cancers is significantly poor when they dedifferentiate or metastasise. Recent studies in glycobiology have shown truncated *O*-glycans to be a hallmark of cancer cells, and when expressed, increase the oncogenicity by promoting dedifferentiation, risk of metastasis by impaired adhesion (mediated by selectins and integrins), and resistance to immunological killing by NK cells. Insight into these truncated glycans provides a complimentary and attractive route for cancer antigen discovery. The recent emergence of immunotherapies against cancers is predicted to harness the potential of using such agents against cancer-associated truncated glycans. In this review, we explore the role of truncated *O*-glycans in cancer progression and metastasis along with some recent studies on the role of *O*-glycans in endocrine cancers affecting the thyroid and adrenal gland.

## 1. Introduction

Glycosylation is the most common post-translational modification process whereby carbohydrates are commonly linked to proteins to form glycans, which are key to a range of functions, namely, cell communication, interaction and adhesion. Glycans can be *N*-linked, where the carbohydrate is linked to the asparagine end of the polypeptide chain, or *O*-linked, where they are covalently linked to serine/threonine. The biosynthesis of *O*-glycans, especially the predominant *N*-acetylgalactosamine (GalNAc), takes place in the Golgi bodies, unlike *N*-glycans synthesis, which is initiated in the endoplasmic reticulum and terminated in the Golgi bodies. Another *O*-glycan that undergoes post-translational modification is *O–N*-acetylglucosamine (*O*-GlcNAc), which is a single monosaccharide seen on nuclear and membrane proteins, and catalysed by a single transferase enzyme unlike the *O*-GalNAc glycans [1,2]. In contrast, the process of *O*-GalNAc glycan synthesis is initiated by a family of about 20 transferase enzymes using the GalNAc backbone to form core glycans, which are further elongated to form various other glycan structures [3].

The process of glycosylation is very well regulated in non-pathological states and is crucial in the normal functioning of the cell along with the various other intercellular interactions [4]. However, the process of glycosylation can be altered in pathological states such as diseases and human cancers [5,6]. In this review, we briefly review the synthesis of truncated *O*-glycans and their role in tumour progression and metastasis. We also highlight a few studies that reported on truncated *O*-glycans as a prognostic indicator in endocrine cancers affecting the thyroid and adrenal glands. We searched PUBMED for articles published in English from 1995 until 2022 using the keywords “*O*-glycans”, “Tn antigen”, “T antigen”, “sialyl Tn antigen”, “ST antigen”, “cancer”, “metastasis’, “endocrine”, “thyroid”, “parathyroid” and “adrenal tumours”. Articles focusing on *N*-glycans and non-English articles were excluded. 

Comprehensive reference to glycobiology and *O*-glycans can be found in the e-textbook *Essentials of Glycobiology, 4th Edition* edited by Ajit Varki, Richard D. Cummings, Jeffrey D. Esko, Pamela Stanley, Gerald W. Hart, Markus Aebi, Debra Mohnen, Taroh Kinoshita, Nicolle H. Packer, James H. Prestegard, Ronald L. Schnaar, and Peter H. Seeberger. Cold Spring Harbor (NY): Cold Spring Harbor Laboratory Press, 2022.

## 2. Synthesis of Truncated *O*-Glycans

*O*-GalNAc glycans are very heterogenous in structure and formed from eight core structures (core 1 to core 8) that exist in mammals. Mucins are *O*-glycosylated proteins that are membrane-bound or exist as secreted forms. They are abundantly distributed on the cell surfaces of the digestive respiratory and urinary tracts. Not all of the core glycans are expressed in all the tissues, with some glycans being tissue specific. The core 1 and 2 glycans are mucin-type glycans seen predominantly in the intestinal tract and mammary tissue, core 3 glycans are seen in intestinal mucosa and salivary glands and core 4 glycans are found in intestinal mucosa. In cancer states, truncated forms such as the Tn antigen (Thomsen nouveau; GalNAcα-Ser/Thr), sialylated STn-antigen (sialyl-Tn; Neu5Acα2–6GalNAcα-Ser/Thr) and the elongated T-antigen (Thomsen–Friedenreich; Galβ1-3GalNAcα-Ser/Thr) are observed [7,8]. The sialylated T antigen (ST; Siaα2,3Galβ1,3GalNAc) is the most common glycan seen in cancers of the breast and stomach [9,10]. These cancer-associated glycans were first described in breast cancer in 1975 [11]. The basic structure similar in all the *O*-GalNAc glycans is the Tn antigen, which may be further sialylated or elongated to form the various core structures. The predominance of a particular glycan synthesised is based on the availability of the specific enzymes present in the Golgi apparatus that regulates the process. The synthesis of the Tn epitope along with the core glycans is shown in Figure 1. 

The enzymes regulating the first step in the synthesis of the *O*-GalNAc glycans are the polypeptide GalNAc- transferases (*pp*GalNAcTs) that determine the glycan-binding site on the serine or threonine amino acid [12]. As many as about 20 different *pp*GalNAcTs have been identified that lead to the modification of the Ser/Thr residues [13,14]. The synthesised Tn can be expressed as a single antigen or in a multivalent form on a polypeptide chain [15]. The Tn antigen is further converted to T antigen catalysed by the enzyme core 1 β3-Gal-transferase (core 1 GalT), also known as *T-synthase*. The core 1 T antigen can undergo a series of further elongations to core 2 structures or be sialylated to form Sialyl Tn catalysed by the enzymes α3-sialyltransferases (ST3Gal) or by α6-sialyltransferases (ST6GalNAc) [16]. One of the key regulators of the function of the T-synthase is the molecular chaperone *COSMC* (gene located on X-chromosome), situated in the endoplasmic reticulum. *COSMC* has important functions such as the prevention of misfolding and degradation of the enzyme T-synthase [15]. Studies in human cancers have shown that aberration of expression of *COSMC* or synthesis of the enzyme T-synthase leads to increased expression of Tn and Sialyl Tn [17,18]. In cancer cells that lack *COSMC* and T-synthase, aberrant glycans are synthesised on the cell surface, which alters the intercellular dynamics, including that of cell recognition by glycan-binding proteins (GBPs) [15]. The variations in glycan core synthesis and capping are based on the roles that the individual glycans play in the interaction, recognition, and immune modulation of cells. Moreover, enzymes that regulate glycan synthesis vary with cell type and cell differentiation [19].

Another truncated glycan, Sialyl-Tn antigen, also known as sTn, contains sialic acid α-2,6 linked to GalNAc. The expression of the enzyme sialyltransferases ST6GalNAc1 is key in the synthesis of sTn along with mutations of the COSMC gene [20]. A range of functions have been elucidated by the linkage of sTn to glycoproteins such as MUC1, CD44, and beta integrin, which are believed to play a role in cell adhesion, migration, and inflammation by augmenting integrin-linked kinase (ILK) and focal adhesion kinase (FAK) mediated cell signaling [18,21]. Munkley et al., using high-grade prostate cancer cells, showed that ST6GalNAc1 induces a switch to a mesenchymal-like pattern with changes in the expression of E-cadherin to N-cadherin gene expression, and increased expression of vimentin, SNAIL, β-microglobulin and β-catenin [22]. The same study showed that the overexpression of ST6GalNAc1 reduced tumorigenicity and metastasis, except for in breast and gastric cancer, where increased expression was associated with more metastatic phenomena [23,24]. This illustrates the fact that the effect of ST6GalNAc1 expression varies with cancer types, and this may be due to the fact that in some cancers the expression of ST6GalNAc1 overrides the expression of T synthase to form the STn antigen and prevents the further addition of sugar residues by the GalNAcT’s enzymes [23].

Core 2 *O*-GalNAc glycans are formed by the addition of the GlcNAc β1-6 branch to the GalNAc residue catalysed by the core 2 β1-6 N-acetylglucosaminyltransferases 1, 2, and 3 (C2GnT-1, C2GnT-3, and C2GnT-4) [25]. Unlike the core 1 *O*-GalNAc glycans, which are ubiquitously distributed, core 2 glycans tend to be synthesised in specific cell types. Whether a core 2 or a truncated *O*-GalNAc glycan is formed is determined by the competitive role played by the C2GnT and STGalNAc enzymes, with the dominant STGalNAc enzymes capping T-antigen with sialic acid and preventing the extension of core 1 glycan [10,26]. The core 2 *O*-GalNAc glycans can undergo further elongation and sialylation to form structures such as Lewis and sialyl Lewis antigens, which have been shown to be highly expressed in many human cancers [27,28,29,30,31,32]. Tumour cells that express C2GnT and harbour core 2 *O*-GalNAc glycans have been shown to evade immune attacks by NK cells and promote metastasis [33]. 

## 3. Role Played by Glycans in Cancer Cell Adhesion

In the progression of cancer, tumour cells acquire features that help in migration and dissemination in the bloodstream, extravasation at distant sites and colonization to form metastatic foci, all mediated by cell adhesion molecules (CAMs). Glycans play a role in cell–cell adhesion and interaction mediated by specific receptors that bind to various glycan epitopes. Some of the glycan-binding receptors shown to play a role in cell adhesion both in the normal function of the immune systems and in tumour cell metastasis include E-selectins, scavenger receptor C-type lectins, galectins and Siglecs. The ligands linked to the glycans when aberrantly expressed bring about changes in signaling, gene expression and cellular interactions that are responsible for the initiation and progression of cancer, promoting aggressive features and metastatic states [34,35,36,37]. 

Selectins are glycoprotein molecules that interact with glycans to mediate various biological processes, especially the adhesion cascade, whereby the cancer cells tether and roll on the endothelium. The family of selectins was classified based on the original cell type where it was identified: E-selectin (endothelium), P-selectin (platelets) and L-selectin (leukocytes) [38]. The binding of aberrant glycans to the E-selectins confers them a distinct advantage in cell migration and metastasis. Knockout studies in mice where the cancer cells did not express the E-selectins showed a reduced ability to metastasise [39]. Studies using human cancer cells derived from the digestive systems showed that the ligands for the selectins seen on the endothelium were sialylated Lewis^x^ and Lewis^a^ antigens [40,41].

An important glycoprotein that promotes interaction at the adherens junction between cells is E-cadherin, and it maintains the integrity and polarity of the epithelial membrane. E-cadherin interaction with galectin-3 mediated via β-catenin is involved in the regulation of the Wnt/β-catenin signalling pathway [42]. Loss of E-cadherin expression has been shown to promote invasion and metastasis, and this loss of expression of E-cadherin has been shown to be due to the upregulation of SNAIL, SLUG, SIP1 and Zeb 1 (repressors that target the promoter of E-cadherin) [43]. Both germline and somatic mutations leading to the truncation of E-cadherin have been observed in gastric cancer [44], breast cancer [45,46] and pancreatic cancer [47]. The loss of E-cadherin also has a role to play in epithelial–mesenchymal transition (EMT), whereby the cell adhesions become destabilised, leading to invasive behaviour and metastasis, which can happen with a single cell or a clone of cells [48]. 

The interaction between the cells and cell matrix, whereby the cells acquire a migratory or metastatic status, is mediated by the crosstalk between E-cadherins and beta integrins, another cell surface glycoprotein [49]. In addition, E-cadherins, the upregulation of N-cadherins by galectin-3, results in adhesion and invasion of cancer cells through the stroma [50]. Other selectins, such as P-selectin (platelets derived) and L-selectins (leukocyte expressed), are also involved in cancer cell adhesion mechanisms and metastasis, but the carbohydrate ligands are not the same as seen in association with E-selectins. The ligand for P-selectin is PSGL-1, which is seen in leukocytes [51]. The lack of PSGL-1 in many human epithelial cancers probably explains why P-selectins play only a minor role in cancer progression. The ligand for L-selectin is sialyl 6-sulfo Lewisx, which is predominantly expressed in normal epithelial cells but is suppressed in malignant cell types [52,53].

Another glycan-binding receptor is C-type lectins (a transmembrane receptor), seen on endothelial cells and myeloid cells that sense cell death [54]. They bear a carbohydrate-binding domain that selectively binds to glycans, mainly Lewis^x^ and Lewis^a^, similar to selectins [55]. The exact role played by C-type lectins in cell adhesion is not very well understood, but a study in breast cancer showed that these lectins may have a role in metastasis like that propagated by selectins [56]. Galectins play a role in endothelial cell adhesion by causing the cells to link to each other and the extracellular matrix (ECM). They interact with cell adhesion molecules (CAMs) such as cadherins, catenins, integrins and TF antigens to induce signals for cancer cell migration and metastasis [57]. High levels of galectin-3 cause endocytosis of integrins resulting in cytoskeletal reorganization, such that the cell adhesions tend to be loose and cause dissemination of the cancer cells [58,59]. Studies in colorectal, glioblastoma, and ovarian cancer have shown that the expression of the C-lectins by the cancer cells detectable by the lectin *Helix pomatia* agglutinin (HPA) is associated with disease progression and poor prognosis [60,61,62].

MUC1 is a transmembrane *O*-glycosylated glycoprotein that also plays an important role in cell adhesion by shielding the small CAMs and inhibiting cancer cell interaction with adjacent cells. Galectin-3 has been shown to co-express with MUC1 and promote the adhesion of the cancer cells to endothelium by exposing the small CAMs that are normally shielded. When MUC1 is overexpressed in cancer along with the TF antigen, the smaller CAMs are exposed, which enhances the motility of the cancer cells and helps them migrate through the basement membrane [63,64]. The expression of MUC1 has been shown to be significantly enhanced in cancers and is associated with metastatic potential and poor prognosis [65]. EGFR bound to Galectin-3 on the cell surface can be associated with the MUC1 extracellular domain to activate the PI3K/AKT pathway to increase mitosis, apoptosis and metastasis [66].

## 4. Glycans and Metastasis

Cancer metastasis is associated with poor prognosis, and the process of dissemination of the disease may be via the lymphatic channels (as seen in the case of melanoma, papillary thyroid cancer and breast cancer) or via the bloodstream (e.g., lung cancer, prostate cancer, and colon cancer). Whatever the mechanism of spread, there appear to be certain steps linked to the metastatic process which may be facilitated by aberrant glycosylation, as shown in Figure 2. Briefly, there is a lack of cohesion between cancer cells, or between the cancer cells and matrix, which enables cancer cells to migrate through the basement lamina and into the vascular channel. Following the entry into the vascular channel, the cancer cells migrate and settle at a distant site. This mechanism is mediated via the interactions between the glycans and molecules, which act as ligands such as cadherins, integrins, selectins and immunoglobulins.

There is evidence that malignant tumours express truncated core-2 glycans that effectively result in metastasis [33,67,68], with other glycan products such as the core-3 and core-4 glycans (*O*-mannosyl glycans) acting as suppressors of metastasis [69,70]. The sialylated core-2 glycans such sialyl Lewis antigens interact with the ligand selectin (expressed by platelets), which are transmembrane glycoprotein cell adhesion molecules and contribute to the systemic circulation of the cancer cells. When migrating through the circulation, the cancer cells adhere to the endothelium by expressing E-selectin, which recognises the glycan Lewis antigens, and then *O*-glycans (mainly the T-antigen and sialyl T-antigen) help tether the cancer cells to the endothelium [71]. The endothelial adhesion of the cancer cells is also mediated by the recognition of galectin-3 by causing homotypic aggregation and promoting metastasis [72]. An interesting observation using breast cancer cell lines was the fact that the aggregation of Tn in the lamellopodia of migrating cells was associated with increased motility and invasion because of GalNAc-T subcellular localization [73].

Another mechanism by which the tumour cells express the truncated glycans is by evasion of natural killer (NK) cell immunity. Using bladder cancer cells in an SCID mouse model, Tsuboi et al. showed that tumour cells that carried the truncated glycans were able to evade the NK cell attack [74]. In normal conditions, NK cells are activated by the interaction between the receptor natural killer group 2 member D (NKG2D) and the tumour cell-expressed ligand MHC Class I-related chain A (MICA), which induces apoptosis [75]. In tumour cells that evade the NK cells, galectin-3 binds to poly-N-acetyllactosamine in the NKG2D-binding site of MICA, thereby preventing the interaction of MICA with NK2GD, and the apoptotic mechanisms are not triggered to promote longer circulation of the tumour cells [70,75]. This mechanism may explain the metastatic mechanism of cancer progression in human cancers. 

## 5. Role of Truncated Glycans in Immune Modulation in Cancer

An important mechanism by which cancer cells remain ‘immortal” is by evasion via immune-mediated cell killing, and this protection is offered to cancer cells by mucins [76]. A group of lectins such as the C-type lectin mannose receptor and Sialic acid-binding immunoglobulin-type lectins (Siglecs) mediate the interaction between the immune system and truncated glycans such as STn seen on cancer cells [61,77]. Another lectin that plays a role in immune regulation is the human CLR macrophage galactose-type lectin (MGL, CD301), which recognises terminal GalNAc moieties such as Tn and STn [78]. High levels of MGL on tumour-associated macrophages are associated with poor prognosis, as shown in stage III colorectal cancer [79].

In humans, there are about 14 functional Siglecs (Sialic acid-binding immunoglobulin-type lectins) widely expressed in the immune system [77,80]. Some of the Siglecs such as Siglecs 3 and 5 expressed on monocyte-derived dendritic cells (moDCs) use the STn epitope to bind to MUC2 [81]. In tumour-associated macrophages, Siglecs 10 and 15 bind to STn antigen to activate the intracellular ITIMs (immunoreceptor tyrosine-based inhibitory motifs) and upregulation of immune checkpoints such as PDL-1 to cause tumour microenvironment immune suppression [82]. The Siglecs that are expressed on NK cells and tumour-associated macrophages (Siglecs 7, 9, 10 and 15) cause immunosuppression and inhibit NK cell-mediated tumour cytotoxicity [82,83,84]. There is evidence that the upregulation of Siglec ligands decreases the susceptibility to NK cell killing in human cancers [84,85,86]. Similarly, Siglec 9 found on NK cells, B cells and monocytes binds to mucins produced by tumours, inducing immunosuppression [81]. 

## 6. Aberrant *O*-GalNAc Glycosylation in Endocrine Cancers

In a recent study that mapped the expression of truncated glycans in a wide range of epithelial and non-epithelial cancers, high levels of expression of Tn and STn were noted [87]. The interesting finding was that of a varied level of expression of Tn and STn based on the cancer phenotypes and their subtypes, with some subtypes expressing only STn (lung adeno- and squamous carcinomas) and not Tn epitopes [87]. Romel et al. also showed that the expression of Tn and STn was inversely proportional to the tumour grade, suggesting that truncated *O*-glycans play a role in the early stages of cancer development and progression [35,37,87]. A recent systematic review and meta-analysis of STn expression in 987 histological tissues from patients with different cancer types indicated a poor prognosis for those with STn-positive tumours [88]. A summary of truncated *O*-glycan expression in various cancers is shown in Table 1. Cancer cells that express a range of GalNAc glycans detectable by the lectin Helix Pomatia Agglutinin (HPA) tend to show highly aggressive features [89]. The lectin HPA has binding specificity to *O*-glycans that bears GalNAc [90], GlcNAc [91], Tn epitope [92] and blood group antigens [93]. Studies looking into the expression of truncated glycans in endocrine cancers are limited to a handful of studies [36,94,95].

a.Thyroid cancer

Differentiated thyroid cancer (DTC) is the most common endocrine cancer and accounts for nearly 3% of all cancers diagnosed worldwide. The global incidence of thyroid cancer is rising [96], and in Singapore, we showed that the incidence had risen by 220% over the last 20 years, possibly due to increased early detection [97]. Thyroid cancers can be very heterogeneous and represent one of the most variable cancers in presentation to outcomes [98]. On one end of the spectrum, we see disease in an indolent form (early differentiated thyroid cancer) with excellent outcomes when diagnosed and treated early, whereas, at the other end, we see disease in the form of poorly differentiated thyroid cancer (PDC) where metastasis is quite common, and then we have the most lethal of all human cancers, anaplastic thyroid cancer (ATC), where patients barely survive beyond 6 months to 1 year despite multimodal therapy. Thyroid cancer progression follows the multistep carcinogenesis theory [99].

In a study, HPA lectin immunohistochemistry was performed using 110 paraffin wax-embedded specimens of benign and malignant thyroid tumours, which showed differential labelling patterns based on the tumour phenotype [94]. In the study, there was a significant difference between HPA binding glycoproteins in benign and malignant thyroid tumours, in that papillary and follicular thyroid cancers expressed more positive labelling localised to the cytoplasm in granular pattern, with marked cell surface localization similar to that previously reported in breast cancer cells [100]. In the same study, lectin affinity chromatography and Western blotting were performed on 128 fresh thyroid samples (46 normal thyroids, 22 goiters, 36 adenomas, 4 follicular cancers, 17 papillary cancers, and 3 metastatic papillary thyroid cancers) and showed a qualitative difference in HPA binding glycoproteins between the benign and malignant phenotypes [94]. The study showed that as the phenotype changed from a benign to a malignant phenotype, truncated glycan expression detectable by the lectin HPA increased, highlighting the fact that truncated glycans cause disease progression. The study also showed positive HPA labelling, indicating that truncated glycan expression is associated with poor survival [94].

In another study that specifically looked at subtypes of follicular thyroid tumours, using HPA lectin histochemistry on archival paraffin wax-embedded specimens of 6 follicular adenomas, 10 minimally invasive follicular carcinomas, 13 widely invasive follicular cancers and 4 metastatic follicular thyroid cancers, the characteristic feature was that as the phenotype of the thyroid tumours changed from a benign to the malignant phenotype, the truncated glycan expressions changed [95]. Another interesting finding was that as the cancers became more aggressive and metastatic, the HPA labelling increased and labelling was positive in tumours with vascular invasion in comparison to capsular invasion only [95].

b.Adrenal cancer

Adrenocortical cancer (ACC) is a rare endocrine tumour arising from the adrenal gland with an incidence of about 1–2 per million population [101,102]. The condition is commonly seen in the fourth to sixth decade of life and is more commonly seen in women [103,104]. About a third of ACCs secrete cortisol and generally have a poor prognosis [101], with a five-year mortality of nearly 80% [36,105]. Nearly half of ACCs, especially in children, have a genetic predisposition such as Li Fraumeni syndrome or multiple endocrine neoplasia 1 (MEN1) [101].

A two-centre study of 32 patients treated with surgery for adrenocortical carcinoma that was evaluated with lectin histochemistry of the cancers excised showed that expression of HPA binding truncated *O*-glycans was associated with invasive characteristics and poorer survival (Figure 3) [36]. Patients with positive HPA labelling had a mean survival of 22 months with a mortality rate of 84% versus negative HPA labelling with a mortality rate of 23% [36]. Labelling was seen along the cell surface and cytoplasm in the cancer cells, similar to that observed in thyroid and breast cancer [94,100] (Figure 4). In the study, positive HPA labelling did not correlate with metastasis, unlike the studies in breast and colon cancer, where the HPA binding partners of integrin α5 and α6 and annexin 2 and 4 were found to play a key role [106,107]. The exact binding partners were not identified in the ACC study as it required characterization using affinity chromatography and mass spectrometry using fresh tumour samples that were not available for the cohort. 

**Table 1 cancers-15-03266-t001:** Studies that evaluated truncated *O*-glycan expression in various human cancers.

Human Cancer Type	% of Truncated Glycans Expressed by Cancers	References
Breast cancer	Tn (57–92%), STn (33–42%), T (32%)	[87,108,109,110]
Gastric cancer	STn (50–78%), Tn (92%), T (20%)	[111,112]
Colorectal cancer	Tn (51–98%), STn (75–87%), T (20%)	[87,113,114]
Lung cancer	Tn (16%), STn (33%), T (10%)	[87]
Pancreatic cancer	Tn (53%), STn (56–97%), T (16–48%)	[87,115]
Skin cancer	Tn (33–50%), STn (24%), T (25%)	[87,116]
Brain cancer	Tn (51%), STn (80%), T (20%)	[87]
Mesenchymal cancers	No expression	[87]

## 7. Conclusions and Future Perspectives

Healthy epithelial cells do not express truncated *O*-GalNAc glycans, and the increased expression of truncated *O*-GalNAc glycans is an indication of cancer. This has been shown in many human cancers, including those of the breast, colon, lung, pancreas, stomach, thyroid and adrenal gland. The truncated *O*-glycans seen are commonly the Tn, STn, T and ST epitope as a result of increased activity of GalNAc-T’s, T-synthase, ST6GalNAc1 and ST3GalNAc1. In addition to playing a role in tumour progression and dedifferentiation, the cancer cells expressing the truncated glycans also develop features to evade immune-mediated killing. The cancer cells also disseminate through the circulation using ligands that bind to the truncated glycans and metastasise and embed in distant tissues. The increased expression of truncated *O*-GalNAc glycans has been shown to be associated with poor prognosis and survival. Only two studies investigated the expression of truncated glycans in endocrine cancers with patterns of expression seen in other human cancers, but the exact mechanisms that the enzymes play in truncated glycan synthesis and mediators of disease progression such as integrins, Siglecs and sialylated Lewis antigens need to be studied further. Studying and characterizing the truncated *O*-glycan profiles not only of the high-risk thyroid cancer subtypes but also other aggressive endocrine cancers as well provides an opportunity to act as a clinical predictor of aggressive disease, aid in accurate treatment recommendations, act as a new biomarker to help in early diagnosis and develop new glycan-based therapies for better control of aggressive endocrine cancers.

## Figures and Tables

**Figure 1 cancers-15-03266-f001:**
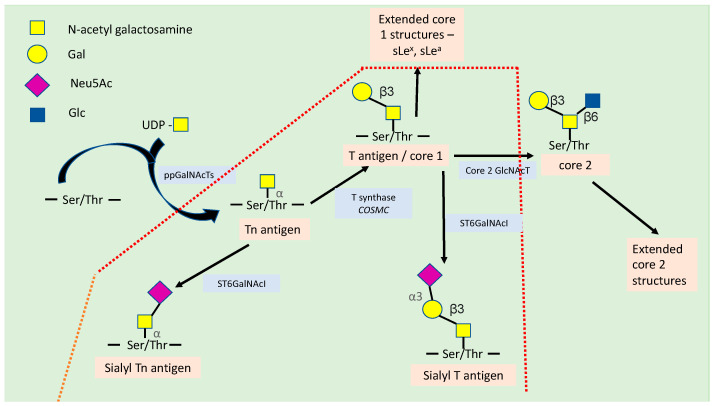
Synthesis of truncated *O*-GalNAc glycans (Tn, T, sialyl T and sialyl Tn antigens). The synthesis begins with attachment of GalNAc to serine or threonine to form Tn, which can be sialylated to form STn or extended to form core 1 glycan. The core 1 glycans then undergo sialylation or are converted to core glycan or form Lewis antigens. Abbreviations: ppGalNAcTs—polypeptide GalNAc- transferases; core 2 GlcNAc T—β1,6 N-Acetylglucosaminyltransferase; Ser/Thr—serine/threonine; UDP—uridine diphosphate; sLe^x^—sialyl Lewis antigen x; sLe^a^—sialyl Lewis antigen a. The truncated core *O*-glycans commonly seen in cancers are boxed in red.

**Figure 2 cancers-15-03266-f002:**
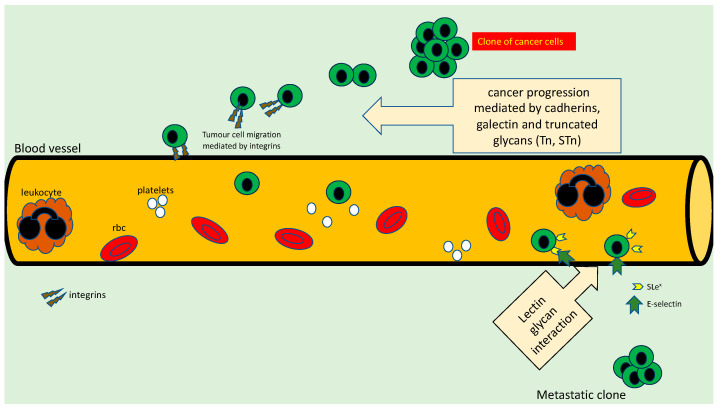
Role of various truncated *O*-glycans in cancer progression and metastasis. Cancer cells lose their adhesive mechanisms between cells, between cells and matrix to break free, and migrate through the extracellular membrane. The cancer cells enter the vascular channels and evade the immune system attack to survive and migrate through the systemic circulation. Using lectin glycan interactions mediated by selectins and sialyl Lewis antigens, the cancer cells extravasate through the endothelium and basement membrane and proliferate at a distant site.

**Figure 3 cancers-15-03266-f003:**
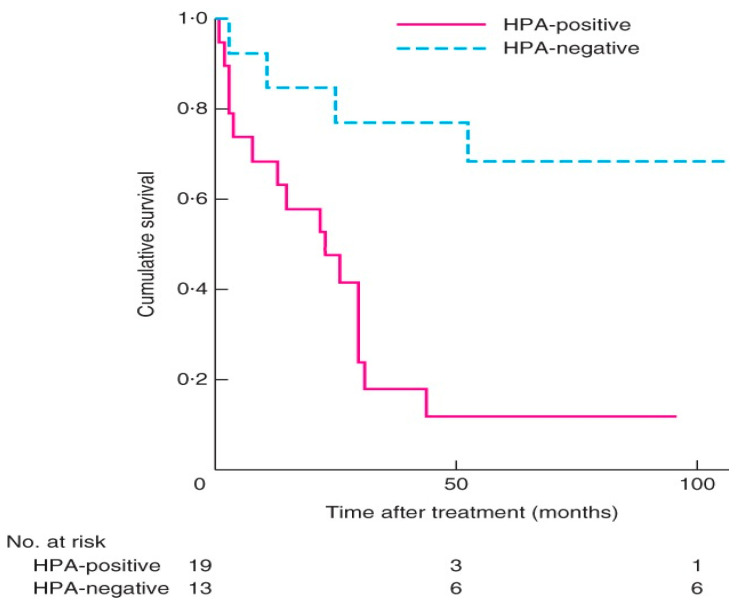
Kaplan–Meier survival curves in patients with adrenocortical carcinoma with positive and negative Helix pomatia agglutinin (HPA) immunolabelling. *p* = 0.002 (log rank test). Used with permission from Parameswaran et al. [36].

**Figure 4 cancers-15-03266-f004:**
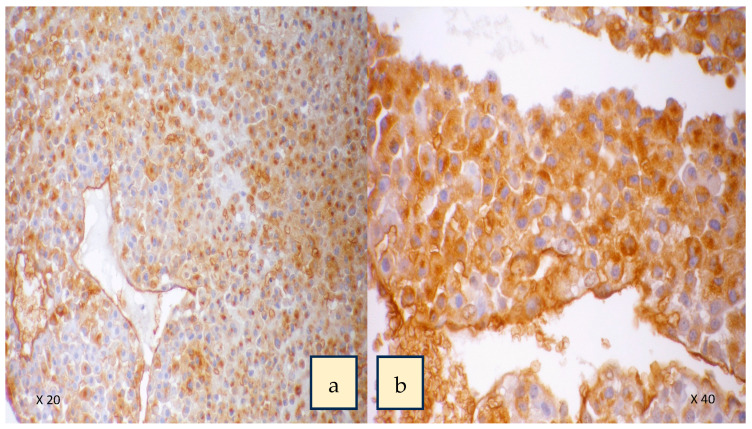
HPA labelling showing intense staining in cytoplasm and cell surface of the cancer cells in adrenocortical carcinoma. Panel (**a**) high-power 20× magnification. Panel (**b**) 40× magnification. Used with permission from Parameswaran et al. [36].

## Data Availability

Data are available in the public domain.

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
