# Peer review of "Role of Truncated O-GalNAc Glycans in Cancer Progression and Metastasis in Endocrine Cancers"

_cancers, 2023, doi:10.3390/cancers15133266_

Round 1

Reviewer 1 Report

Many thanks for your efforts in putting this review together. Broadly, from my perspective, it is a good overview of where we currently are as a community. It lacks some depth in places, and is prone to generalisation, but this is not necessarily a negative as it improves readability and keeps the focus. However, I have a few comments which I feel would improve the piece; please see below. NB. There are few language issues throughout – missing words, missing ‘s’s etc – nothing major as I could easily understand, but if it can be improved, that would be wonderful. 

At the bottom of the introduction, it would be good to include something along the lines of: ‘for further information on O-linked glycosylation please refer to the open access textbook Fundamentals of Glycobiology’ (ref). This would allow interested readers to follow up in greater depth if desired.

“However, alterations in the process of glycosylation can lead to pathological states such as diseases and human cancers [2, 3].” This statement needs tweaking as it suggests that changes in glycosylation lead directly to cancers – plus cancers are diseases.

It is a shame that the ST antigen or diST antigen were missed from the search terms – but I am not proposing to you bring these in unless you wish. I am a little confused – did you manually go through all pubmed articles featuring O-linked glycans from 1995-2022? This would be 1934 articles? This is too much, in which case can you explain what combinations of search terms you used? 

“O-glycans are very heterogenous in structure due to the combinations of the 8 core- structures (core 1 to core 8) that exists in mammals.” You need to state something along the lines of tissue specificity – ie not all tissues express all these core structures – eg the colon, core 3, the breast, core 2. Otherwise people think the system is extremely random. 

Just below this sentence – very odd – you mention Tn, STn and T and then discuss breast cancers, but miss the dominant (80-95%; depending on group) glycan – the ST structure, which is featured in your diagram. Optional, but would improve to include the disialylT structure in the diagram at least (you likely not find much out there for the text). 

“These cancer associated antigens were first described in breast cancer in 1975, and has been shown to have immunogenic properties [6].” Lovely reference and good to include but, in the context of the paragraph, you are suggesting that these specific structures were known about in 1975 which isn’t the case – not 100% sure this reference covers you for ‘immunogenic properties’ either. Needless to say you are broadly correct in what you say – it’s just this ref is carrying a lot of weight here!

It looks as if your diagram has an extra 3 in ST6GalNAc-1 and ST3Gal-1 – also lacking the ‘NAc’ – please use the more commonly used forms throughout – or used the gene names as well so people don’t get confused. 

Can you change ‘the truncated core O-linked glycans seen in cancers are boxed in red’ to ‘…commonly seen in cancers…’ only some cancers carry longer O-linked glycans. 

I like the focus on the editor’s work! Very canny!!

Section on E-cadherin is a little lost in that although it is a glycoprotein – most proteins are – we do not yet know the specifics about the impact of differentiation O-linked glycosylation. If you wanted to highlight a cancer associated protein where we know about the impacts of differential glycosylation, EGFR would have been better. But it’s fine to leave as it is. 

“An important mechanism by which cancer cells remain ‘immortal” is by evading by immune-mediated cell killing and this protection offered to cancer cells is by mucins which are essentially O-glycans.” This sentence has me worried…. You really cannot say mucins are essentially O-glycans?! 

The tumour cells can induce immune suppression by releasing anti-inflammatory cytokines and decreasing chemokine CCL3 when CD206 (C-type lectin mannose receptor) on tumour associated macrophages binds with the sialyl-Tn domains and MUC 16 [40]. Please re-read the paper and correct this statement. 

Very odd mucin – siglec section – no mention of the ST glycan. No mention of Varki / Bertozzi / Burchell / Van Kooyk / Laubli papers. Should be here as one of the reasons Carolyn won the Nobel. 

I’m not 100% sure about using already published images in a review – never seen this before – but if you have permission and have referenced, I guess this is okay – probably best to check with the journals in which they were published too as this may be copyright infringement. 

Much better section on the specific cancers. 

“Normal human cells do not express truncated O-glycans and increased expression of truncated O-glycans is a feature of cancer cells.” I’m sorry this is simply not true – please read around glycans on T cells in different states of activation or non-cancerous diseases such as IPF… or simply change this to reflect the fact that healthy epithelial cells do not usually express truncated O-linked glycans and when they do it is frequently an indication of cancer. 

Again the missing of ST is strange – eg in breast cancers approx. 25% are STn whilst approx. 85% are ST. Tn cancer data always debated as we don’t know how much is on surface – but I don’t expect you to go into this. 

Needs some improving - just run it past a native speaker - should take them about an hour. 

Author Response

Reviewer 1

We thank the reviewer for the valuable and insightful comments. We have addressed almost all of the concerns raised and highlighted these responses in the manuscript.

Comments and Suggestions for Authors

Many thanks for your efforts in putting this review together. Broadly, from my perspective, it is a good overview of where we currently are as a community. It lacks some depth in places, and is prone to generalisation, but this is not necessarily a negative as it improves readability and keeps the focus. However, I have a few comments which I feel would improve the piece; please see below. NB. There are few language issues throughout – missing words, missing ‘s’s etc – nothing major as I could easily understand, but if it can be improved, that would be wonderful. 

At the bottom of the introduction, it would be good to include something along the lines of: ‘for further information on O-linked glycosylation please refer to the open access textbook Fundamentals of Glycobiology’ (ref). This would allow interested readers to follow up in greater depth if desired.

Added a reference to the textbook

“However, alterations in the process of glycosylation can lead to pathological states such as diseases and human cancers [2, 3].” This statement needs tweaking as it suggests that changes in glycosylation lead directly to cancers – plus cancers are diseases.

Agree with the reviewer and have amended the statement

It is a shame that the ST antigen or diST antigen were missed from the search terms – but I am not proposing to you bring these in unless you wish. I am a little confused – did you manually go through all pubmed articles featuring O-linked glycans from 1995-2022? This would be 1934 articles? This is too much, in which case can you explain what combinations of search terms you used? 

We focussed on articles that discussed O-glycans and association with cancers and retrieved 498 articles.

“O-glycans are very heterogenous in structure due to the combinations of the 8 core- structures (core 1 to core 8) that exists in mammals.” You need to state something along the lines of tissue specificity – ie not all tissues express all these core structures – eg the colon, core 3, the breast, core 2. Otherwise people think the system is extremely random. 

The reviewer has raised a good point and this has been added to the introduction section.

Just below this sentence – very odd – you mention Tn, STn and T and then discuss breast cancers, but miss the dominant (80-95%; depending on group) glycan – the ST structure, which is featured in your diagram. Optional, but would improve to include the disialylT structure in the diagram at least (you likely not find much out there for the text). 

Have included a statement about ST antigen

“These cancer associated antigens were first described in breast cancer in 1975, and has been shown to have immunogenic properties [6].” Lovely reference and good to include but, in the context of the paragraph, you are suggesting that these specific structures were known about in 1975 which isn’t the case – not 100% sure this reference covers you for ‘immunogenic properties’ either. Needless to say you are broadly correct in what you say – it’s just this ref is carrying a lot of weight here!

Have amended the statement.

It looks as if your diagram has an extra 3 in ST6GalNAc-1 and ST3Gal-1 – also lacking the ‘NAc’ – please use the more commonly used forms throughout – or used the gene names as well so people don’t get confused. 

Amended in the figure

Can you change ‘the truncated core O-linked glycans seen in cancers are boxed in red’ to ‘…commonly seen in cancers…’ only some cancers carry longer O-linked glycans. 

amended

I like the focus on the editor’s work! Very canny!!

Section on E-cadherin is a little lost in that although it is a glycoprotein – most proteins are – we do not yet know the specifics about the impact of differentiation O-linked glycosylation. If you wanted to highlight a cancer associated protein where we know about the impacts of differential glycosylation, EGFR would have been better. But it’s fine to leave as it is. 

Thank you for the comments. Have made some changes.

“An important mechanism by which cancer cells remain ‘immortal” is by evading by immune-mediated cell killing and this protection offered to cancer cells is by mucins which are essentially O-glycans.” This sentence has me worried…. You really cannot say mucins are essentially O-glycans?! 

Rightly pointed out and have amended the statement.

The tumour cells can induce immune suppression by releasing anti-inflammatory cytokines and decreasing chemokine CCL3 when CD206 (C-type lectin mannose receptor) on tumour associated macrophages binds with the sialyl-Tn domains and MUC 16 [40]. Please re-read the paper and correct this statement. 

We agree that the statement did not read well. Amended the section.

Very odd mucin – siglec section – no mention of the ST glycan. No mention of Varki / Bertozzi / Burchell / Van Kooyk / Laubli papers. Should be here as one of the reasons Carolyn won the Nobel. 

Have referenced Bertozzi and Varki

I’m not 100% sure about using already published images in a review – never seen this before – but if you have permission and have referenced, I guess this is okay – probably best to check with the journals in which they were published too as this may be copyright infringement. 

Much better section on the specific cancers. 

“Normal human cells do not express truncated O-glycans and increased expression of truncated O-glycans is a feature of cancer cells.” I’m sorry this is simply not true – please read around glycans on T cells in different states of activation or non-cancerous diseases such as IPF… or simply change this to reflect the fact that healthy epithelial cells do not usually express truncated O-linked glycans and when they do it is frequently an indication of cancer. 

Amended the sentence

Again the missing of ST is strange – eg in breast cancers approx. 25% are STn whilst approx. 85% are ST. Tn cancer data always debated as we don’t know how much is on surface – but I don’t expect you to go into this. 

ST and STn has not been studied in thyroid cancer and hence omitted from the endocrine section. We are currently evaluating the T, Tn, STn and ST antigen expression in thyroid and adrenal cancer.

Reviewer 2 Report

In this manuscript, the authors summarize the roles of truncated O-glycans in cancer progression and metastasis, and some recent studies in endocrine cancers.

Major comments:

1.  At the end of first paragraph in introduction, the authors mentioned “…unlike other glycans such as N-glycans and O-N-acetylglucosamine (O-GlcNAc) etc., which are initiated in the endoplasmic reticulum and terminated in the Golgi bodies.” Please double-check where O-GlcNAc occurs and whether there is elongation of O-GlcNAc to form more complex structures. 

2. Should core 2 glycans be included as the truncated O-glycans seen in cancers? In reference 37, it was reported “upregulation of C2GnT is closely correlated with progression of bladder tumors”. Additionally, there are other references suggesting the role of core2 O-glycans in some cancers, such as prostate cancer, colorectal cancer, bladder cancer, etc. The authors should give some summary about core2 functions. Additionally, in Figure1, the core2 structure should be also boxed in red.

3. In Figure1, should core 3 and core 4 structures be included in the synthesis of truncated O-glycans, although they may not promote cancer progression? In Figure 1, it should be illustrated that what each symbol represents.

4. In the part3 “Role played by truncated glycans in adhesion of cancer cells”, the authors should provide more information about the roles of glycans not only focus on the roles of some proteins. 

5. In the legend of Figure2, the definitions of abbreviations should be listed, such as EMT, TAMs, rbc.  

6. Please double-check grammar, spelling, and punctuation in the manuscript.

Minor comments:

1.     “ppGalNAcT’s” should be changed to “ppGalNAcTs”.

2.     On page3, in the 78th line, “…catalyzed the enzyme core1…” should be “… catalyzed by the enzyme core1…”.

3.     On page3, in the 116th line, “A hallmark of cancer is cells is lack of …” should be “A hallmark of cancer is cells are lack of …”.

4.  On page3, in the 119th line, “Adherens junctions” should be “adherens junctions”.

Please double-check grammar, spelling, and punctuation in the manuscript.

Author Response

We thank the reviewer for the valuable and insightful comments. We have addressed almost all of the concerns raised and highlighted these responses in the manuscript.

In this manuscript, the authors summarize the roles of truncated O-glycans in cancer progression and metastasis, and some recent studies in endocrine cancers.

Major comments:

  1. At the end of first paragraph in introduction, the authors mentioned “…unlike other glycans such as N-glycans and O-N-acetylglucosamine (O-GlcNAc) etc., which are initiated in the endoplasmic reticulum and terminated in the Golgi bodies.” Please double-check where O-GlcNAc occurs and whether there is elongation of O-GlcNAc to form more complex structures.

Have corrected and added a new sentence.

  1. Should core 2 glycans be included as the truncated O-glycans seen in cancers? In reference 37, it was reported “upregulation of C2GnT is closely correlated with progression of bladder tumors”. Additionally, there are other references suggesting the role of core2 O-glycans in some cancers, such as prostate cancer, colorectal cancer, bladder cancer, etc. The authors should give some summary about core2 functions. Additionally, in Figure1, the core2 structure should be also boxed in red.

Have added a paragraph on C2 glycans

  1. In Figure1, should core 3 and core 4 structures be included in the synthesis of truncated O-glycans, although they may not promote cancer progression? In Figure 1, it should be illustrated that what each symbol represents.

Core 3 and 4 are not truncated glycans and hence not included in this review. Have made the necessary changes in figure 1 to illustrate the symbols.

  1. In the part3 “Role played by truncated glycans in adhesion of cancer cells”, the authors should provide more information about the roles of glycans not only focus on the roles of some proteins. 

Have included a few more proteins and their interaction with glycans

  1. In the legend of Figure2, the definitions of abbreviations should be listed, such as EMT, TAMs, rbc.  

Amended in the figure

  1. Please double-check grammar, spelling, and punctuation in the manuscript.

Corrected

Minor comments:

  1. “ppGalNAcT’s” should be changed to “ppGalNAcTs”.

amended

  1. On page3, in the 78thline, “…catalyzed the enzyme core1…” should be “… catalyzed by the enzyme core1…”.

amended

  1. On page3, in the 116thline, “A hallmark of cancer is cells is lack of …” should be “A hallmark of cancer is cells are lack of …”.

amended

  1. On page3, in the 119thline, “Adherens junctions” should be “adherens junctions”.

amended

Reviewer 3 Report

The authors give a review on truncated O-glycans in different cancers. This is a very important topic as the detection of these glycans may help in diagnosis and prognosis.

Due to the fact that the authors were just searching for keywords and do not seem to have experience in the field of glycobiology, the selection of papers is rather weak. There are many more which would fit into such a review. Some references are missplaced (line 77 shuld bei [7]).

The authors do not use the common symbols for glycans in Fig. 1. Please refer to https://www.ncbi.nlm.nih.gov/glycans/snfg.html for correction of this. Also an explanation should be given in the Figure description.

There are many mistakes especially in the chapters 1-3. Singular/plural, prepostions and word order. This makes the paper extremely difficult to understand.

Some examples:

Line 58: O-glycans are very heterogenous in structure due to the combinations (wrong choice of word) of the 8 core-structures that exists in mammals. The most commonly seen O-glycans is are mucins which is are abundandtly distributed in human organ systems.

Line 78: ... converted to T antigen catalyzed by the enzyme ....

Line 116: A hallmark of cancer is cells is lack of adhesion ... ????

Author Response

Reviewer 3

We thank the reviewer for the valuable comments. We have addressed almost all of the concerns raised and highlighted these responses in the manuscript.

The authors give a review on truncated O-glycans in different cancers. This is a very important topic as the detection of these glycans may help in diagnosis and prognosis.

Have added a table of truncated glycan expression in various human cancers.

Due to the fact that the authors were just searching for keywords and do not seem to have experience in the field of glycobiology, the selection of papers is rather weak. There are many more which would fit into such a review. Some references are misplaced (line 77 should be [7]).

Have tried to incorporate a few more papers of relevance.

The authors do not use the common symbols for glycans in Fig. 1. Please refer to https://www.ncbi.nlm.nih.gov/glycans/snfg.html for correction of this. Also an explanation should be given in the Figure description.

Have amended the figures using the correct nomenclature as suggested.

Some examples:

Line 58: O-glycans are very heterogenous in structure due to the combinations (wrong choice of word) of the 8 core-structures that exists in mammals. The most commonly seen O-glycans isare mucins which is are abundandtly distributed in human organ systems. (the grammatical errors have been addressed)

Line 78: ... converted to T antigen catalyzed by the enzyme ....(corrected)

Line 116: A hallmark of cancer is cells is lack of adhesion ... ???? (modified the sentence)

Round 2

Reviewer 2 Report

In the revised manuscript, it was mentioned that “Mucins are the most common O-glycans which can be membrane bound or as secreted forms are abundantly distributed on the cell surfaces of the digestive respiratory and urinary tracts.” Mucins are not O-glycans. They are glycosylated proteins. Again, please double-check the grammars in manuscript.

In the revised manuscript, it was mentioned that “Mucins are the most common O-glycans which can be membrane bound or as secreted forms are abundantly distributed on the cell surfaces of the digestive respiratory and urinary tracts.” Mucins are not O-glycans. They are glycosylated proteins. Again, please double-check the grammars in manuscript.

Author Response

We thank the reviewer for the error made. Have made the necessary correction as requested. 

Reviewer 3 Report

The paper improved a lot by the addition of further references.

Fig. 1: the common symbols for monosaccharides are still wrong. Gal containing monosaccharides (Gal, GalNAc) must be yellow and not brown; sialic acid should be purple and not red. Furthermore, the explanation of these symbols is still missing.

The language is still very week, regarding singular/plural, prepostitions, word order and commas.

External editing is required.

Some examples:

Line 104 enzymes Plural!

Line 105 determines Plural!

Line 118 that which

Line 168 interacts Plural!

Line 170 were was Singular!

Line 176 was were Plural!

Line 203 is are Plural!

Line 204 senses Plural!

Line 295 uses Plural!

Author Response

Reviewer 3

We thank the reviewer for the valuable comments. We have made the necessary corrections in the figure and the manuscript.